# SceneLock: Reversible Adversarial Learning for Camera-Based Autonomous Driving Protection

## Abstract

The advancement of autonomous driving technology hinges on large-scale data collection to train camera-based deep neural network 3D object detectors. However, these valuable datasets are at risk of unauthorized access and misuse by malicious actors, jeopardizing intellectual property, remote deployment, and the privacy of sensitive information captured during data collection. We propose a novel reversible adversarial learning framework, referred to as **SceneLock**, aimed at protecting autonomous driving data from unauthorized use. Our method conducts adversarial perturbations through a carefully designed **N**oise **S**erialization **E**ncoding module (**NSE**), which significantly degrades image quality and renders the data ineffective for unauthorized artificial intelligence models and manual annotation. To ensure legitimate access remains unaffected, we integrate advanced image steganography to embed perturbation values within the images. Furthermore, authorized users can extract these values using appropriate decryption tools through the **N**oise **S**erialization **D**ecoding module (**NSD**) to restore the original high-quality images. Experimental results demonstrate that our approach effectively safeguards data integrity against unauthorized use while maintaining availability for legitimate purposes. This dual-layer protection highlights the potential of our method to enhance data security in the autonomous driving domain.

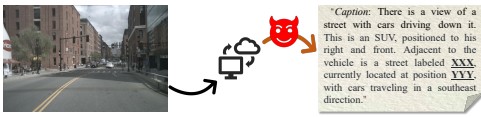 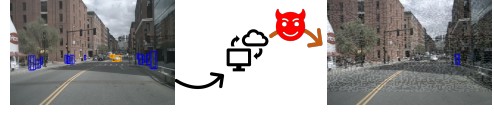

(a) Scene Level: Clean scene data may lead to privacy security breaches.   (b) Object Level: Loss of perception regarding vehicles on the road.

Figure 1: Camera-captured scene data poses security risks during the transition from deployment to service: Scene-level privacy breaches and Object-level perception loss.

## 1 Introduction

Camera-based deep neural network 3D object detectors have demonstrated exceptional performance on multiple large-scale autonomous driving datasets, including benchmark datasets such as KITTI Geiger et al. (2012), nuScenes Caesar et al. (2020), and Waymo Sun et al. (2020). The success of these detectors largely hinges on the collection and utilization of extensive amounts of high-quality data to train and fine-tune complex models. However, the accumulation of large-scale datasets driven by data consensus introduces significant risks of unauthorized access and misuse. Such access may lead to the exposure of Scene-level sensitive information, including images of confidential infrastructure or objects encountered during data collection. Furthermore, it could result in the failure of Object-level perception by models at the deployment stage, as shown in Figure 1.

Traditional data protection methods, such as encryption Lagendijk et al. (2012) and access control Qiu et al. (2020), may not be sufficient to thwart sophisticated adversaries who can bypass security measures or exploit vulnerabilities in AI models. Therefore, there is an urgent need for robust mechanisms that can protect datasets from unauthorized use while preserving their utility for legitimate applications.

Figure 2: Clean scene data is encoded using Noise Serialization Encoding to generate adversarial samples with perturbation noise, which can be decoded back to the original image using Noise Serialization Decoding.

In this paper, we propose a novel reversible adversarial learning framework for the protection of camera-based autonomous driving scenes, as shown in Figure 2. Due to its unique access control mechanism for enabling or disabling, it is termed SceneLock. Our approach is grounded in the following key innovations:

• **Adversarial Perturbations for Data Protection:** Extending adversarial perturbation tasks into data protection, we introduce meticulously designed adversarial perturbations into the collected images, injecting high levels of noise that significantly degrade image quality. These perturbations are intended to impair the performance of unauthorized AI models, preventing the extraction of useful features from the data. Furthermore, the degraded image quality obstructs manual annotation efforts by unauthorized parties.

• **Integration of Image Steganography:** Combining perturbation noise with reversible encoding. To ensure that authorized users can access the original data without degradation, we embed the perturbation values within the images using advanced steganography techniques. Authorized personnel with the appropriate decryption tools can extract these values and restore the images to their pristine state, maintaining the data's integrity for legitimate use.

SceneLock provides robust protection against unauthorized data exploitation while ensuring no loss in data quality or accessibility for legitimate users. Our contributions can be summarized as follows:

• We propose a reversible adversarial learning framework for the protection of camera-based autonomous driving scenes. To our knowledge, this method represents the first application of reversible adversarial perturbations in autonomous driving, laying the foundation for more robust and reliable systems.

• We propose a Noise Serialization Encoding module (NSE) and a Noise Serialization Decoding module (NSD) for the reversible embedding and extraction of perturbation noise. These two modules ensure excellent adversarial performance and high fidelity during data recovery.

• We conduct extensive experiments to validate the effectiveness of our approach in protecting data integrity while maintaining availability for authorized applications.

## 2 RELATED WORK

In this section, we provide an overview of current camera-based 3D perception methods and adversarial perturbation attacks.

**Camera-based 3D Perception.** With the success of deep learning in the visual domain, pure vision-based approaches have emerged as an important branch of autonomous driving perception. FCOS3D Wang et al. (2021a) and PGD-DET Wang et al. (2021b) are monocular 3D detectors that typically rely solely on 2D images from a principal viewpoint for 3D estimation. Despite significant advancements, their performance remains constrained by depth uncertainty and a lack of diverse perspectives in 3D space. In contrast, BEV-based methods effectively address these issues and demonstrate superior performance. BEVDet Huang et al. (2021) introduced the first high-performance BEV detector, employing the Lift-Splat-Shoot (LSS) Philion & Fidler (2020) method to convert panoramic multi-view data into a bird's-eye view. BEVDepth Li et al. (2023b) supervises depth estimation by projecting 3D point clouds onto the image. Notably, BEVDet later incorporated temporal feature fusion, termed BEVDet-4D Huang & Huang (2022). Additionally, query-based Transformer architectures have gained considerable attention in 3D detectors. DETR3D, inspired by DETR, is

Figure 3: The overall framework of the SceneLock.

based on a Transformer backbone and associates 2D features with 3D bounding box predictions through geometric projection. PETR Liu et al. (2022) enhances 2D feature representation by encoding position-aware 3D representations. BEVFormer Li et al. (2022) refines BEV queries using spatial and temporal attention mechanisms. These approaches further unlock the potential of 3D detectors within the Transformer framework while reducing reliance on depth estimation. Furthermore, camera-based 3D occupancy prediction tasks, such as DHD Wu et al. (2024) and FlashOcc Yu et al. (2023), are gradually becoming mainstream in 3D perception.

**Adversarial Perturbation Attack.** In the realm of 2D images, deep neural networks (DNNs) have been recognized as vulnerable to adversarial attacks, which demonstrate significant potential threats and value Modas et al. (2019); Fan et al. (2020); Goodfellow et al. (2014). Carlini & Wagner (C&W) Carlini & Wagner (2017) first proposed generating adversarial examples by adding imperceptible perturbations to the original images, thereby assessing model robustness and leading to highly confident incorrect predictions. This characteristic has also been validated in object detection tasks Liang et al. (2022); Liu et al. (2019); Wu et al. (2020). Conducting adversarial attacks solely at the 2D level is not directly applicable to robustness studies in the 3D physical world. Consequently, recent research has shifted its focus toward 3D perception issues to elucidate potential safety threats in real-world environments Xiang et al. (2019); Wicker & Kwiatkowska (2019); Hamdi et al. (2020). For instance, Adv3D Li et al. (2023a) utilizes NeRF differentiable rendering techniques to synthesize target vehicles within realistic camera scenes, while BEV-Attack Xie et al. (2024) propose using a 3D Surrogate model to learn noise patches that can interfere with the performance of 3D detectors.

In this work, we aim to generate adversarial samples for protecting driving scenarios and achieve recoverability of these samples through decoding. The comparisons in Table 1 indicate that our method exhibits stronger transferability, lower requirements, and greater practical applicability.

# 3 METHOD

## 3.1 OVERVIEW

In this paper, we will present a detailed account of the SceneLock framework for camera-based autonomous driving protection, which employs reversible adversarial learning. The overall structure of SceneLock is depicted in Figure 3. It consists of two main components: Noise Serialization Encoding (NSE) and Noise Serialization Decoding (NSD). In Section 3.2, we will elaborate on the noise serialization encoding process, covering gradient encoding, serialization encoding, and the optional Reversible Data Embedding (RDE) module. In Section 3.3, we will explain how the noise serialization decoding module utilizes specific tools to extract binary codes from embedded images and decode particular image perturbation noise, ultimately recovering the original image by subtracting these noises. Additionally, in Appendix A section, we provide supplementary information on the principles of steganography in image applications to enhance readers' understanding of the related content.

## 3.2 NOISE SERIALIZATION ENCODING

Prior to delving into implicit noise encoding attacks, it is essential to first discuss the limitations of Reversible Data Encoding (RDE) regarding the length of encodable bytes, as detailed in Appendix

A. Our work designs a Serialization Encoding (SE) method, an efficient perturbation compression technique that utilizes superpixels in place of individual pixels. This approach reduces storage requirements by applying gradient smoothing to superpixels while maintaining adversarial efficacy. Consequently, even with reduced data space, the perturbations remain effective in challenging the model.

**Gradient Contribution Map Calculation.** We denote the input clean scene image as $x \in \mathbb{R}^{C \times H \times W}$, the ground truth label as $y^{true}$ (Bounding Box $B$ and Classification Category $Cls$), and the output result of the surrogate model as $y = f_\theta(x)$. Super-pixel size as $h \times w$, $\eta$ represents the perturbation generated by surrogate model. The adversarial examples $x'_{adv}$ can be expressed as:

$$x'_{adv} = \max(0, \min(x + \mathcal{T}(\eta), 1)) \tag{1}$$

where $\mathcal{T}$ is a function designed for dimension expansion and padding. Due to the adoption of super-pixels, $\eta$ effectively functions as a simplified perturbation block with a two-dimensional shape of $(\lfloor H/h \rfloor, \lfloor W/w \rfloor)$. Specifically, $\eta_{ij}$ denotes the perturbation of the super-pixel at position $(i, j)$, while the function $\mathcal{T}$ extends this perturbation to cover the area $(c, i : i + h, j : j + w)$, where ( $0 \leq i \leq \lfloor H/h \rfloor$ ) and ( $0 \leq j \leq \lfloor W/w \rfloor$ ). We utilize $\epsilon$ as the unit of perturbation and employ a three-bit code to represent the magnitude of perturbation at $\eta_{ij}$, which indicates the count of unit perturbations.

Then, we analyze how to construct perturbations based on the deviations of the gradients. Initially, the perturbation gradient is obtained through the 2D surrogate model detector $f_\theta$.

$$f_\theta(x'_{adv}) = y \neq y^{true}, s.t. ||\mathcal{T}(\eta)||_\infty \leq \epsilon \cdot m \tag{2}$$

where $m$ represents the maximum multiplicative factor stored in three bits. To generate $x'_{adv}$, we compute pixel-wise gradients from the loss function and add perturbations to increase the loss in non-targeted attacks. Given the variability of gradient values across different positions, applying uniform perturbations would result in varied impacts on the loss function. Therefore, we prioritize larger perturbations at points that exert a greater influence on the loss function, as these regions are more sensitive to input changes that can significantly affect the final classification decision, thereby enhancing the effectiveness of the attack. Consequently by smoothing the gradients for each super-pixels:

$$\nabla_\eta \mathcal{J}(x, y^{true})_{ij} = \frac{\sum_0^C \sum_0^H \sum_0^W \nabla_x J(x, y^{true})_{ij}}{C \times H \times W} \tag{3}$$

$$\mathcal{A} = |\nabla_\eta \mathcal{J}(x, y^{true})| \tag{4}$$

Note that by calculating $\nabla_\eta \mathcal{J}(x, y^{true})$, an absolute value matrix $\mathcal{A}$ can be obtained. Consequently, based on $\mathcal{A}$, we can construct a gradient score map $\mathcal{E}$ to determine the sensitivity of different block super-pixels:

$$\mathcal{E} = \frac{\exp(\mathcal{A}_{ij})}{\sum_{p=0}^{i \times j} \exp(\mathcal{A}_p)}, s.t. i \in [0, \lfloor H/h \rfloor], j \in [0, \lfloor W/w \rfloor] \tag{5}$$

where $\mathcal{E}$ denotes the impact of perturbations at different positions on the loss function, termed as the contribution score of gradients to the deviation in the loss function. Considering the variations in contributions from different blocks, the generated perturbation values are quantized into multiple levels. The complete algorithmic process is presented as outlined in Algorithm 1.

It is important to note that the NSE is an encoding module specifically designed for scene detection and scene perception tasks. Therefore, it differs from single-class weakly supervised classification attack tasks, which primarily aim to achieve semantic interference and detection failure. To disrupt the confidence of targets in a scene, we can define the following loss function:

$$\mathcal{J}(x, y^{true}) = (-1)_{y_{cls}^{true}} \cdot \log\left(\frac{\exp(f_\theta(x, y_{cls_i}^{true}))}{\sum_{i=1}^N \exp(f_\theta(x, i))}\right) + \sum_{i=1}^N ||f_\theta(x)_b - y_{B_i}^{true}||_1 \tag{6}$$

where $(-1)_{y_{cls}^{true}}$ is the one-hot encoding of $y_{cls}^{true}$, $y_B^{true}$ represents the total number of object boxes and $y_{cls}^{true}$ is the correct category of the image. The detailed NSE attack procedure is outlined in Algorithm 2, providing a step-by-step guide for implementing this attack strategy.

**Complete Perturbation of Noise.** We use the perturbation $\eta$ generated by Algorithm 2 and the clean image $x$ as inputs. A direction $q$ is randomly selected based on the chosen perturbation $\eta_{ij}$. Adding

---

**Algorithm 1** Gradient Score Map Contribution Calculation

---

1: **Input:** Gradient of the super-pixels $\nabla_x J(x, y^{true})_{ij}$
2: **Input:** Percentage $\mathcal{PCT}$; unit perturbation $\epsilon$
3: **Output:** Perturbation $\xi$
4: Initialize the absolution value matrix $\mathcal{A}$, the sign function $\mathcal{S}$ and the contribution score matrix $\mathcal{E}$
5: $\mathcal{A} = |\nabla_\eta \mathcal{J}(x, y^{true})|$
6: $\mathcal{S} = sign(\nabla_\eta \mathcal{J}(x, y^{true}))$
7: Compute the contribution score matrix $\mathcal{E}$
8: **for** $0 \leq i \leq \lfloor H/h \rfloor$ **do**
9:     **for** $0 \leq j \leq \lfloor W/w \rfloor$ **do**
10:         $\mathcal{E} = e^{\mathcal{A}_{ij}} / \sum_{p=0}^{i \times j} e^{\mathcal{A}_p}$
11:     **end for**
12: **end for**
13: Obtain the coordinates of the top $\mathcal{PCT}$ values in the $\mathcal{E}$
14: Set the values at these positions in $\mathcal{A}$ to 2, and the rest to 1
15: Return $\xi$

---

perturbations in the $q$ direction alters the model's confidence $p$. If the direction $q$ fails to decrease $p(y^{true}|_x + \mathcal{T}(\eta + q \cdot \epsilon))$, the direction $q$ is reversed. For each perturbed point, we record its contribution to the reduction in confidence. After a predefined number of perturbation iterations $m$, we identify and select the three points that yield the most significant decrease in model confidence from these records. As the perturbations of superpixel blocks may have reached their maximum threshold, further increasing these perturbations may have limited direct impact on confidence. Therefore, we exclude these points and enhance only those where additional perturbations can be applied.

---

**Algorithm 2** Noise Serialization Encoding

---

1: **Input:** Clean image $x$; and the true boxes label $y^{true}$
2: **Input:** Percentage $\mathcal{PCT}$; unit perturbation $\epsilon$; iteration $\mathcal{I}$; and the maximum multiplicative factor $m$
3: **Output:** Adversarial examples $x'_{adv}$ ; Perturbation $\eta$
4: Initialize $g_0 = 0$, $x'^0_{adv} = x$, $\eta_0 = $ zero matrix
5: **for** $i = 0$ to $\mathcal{I} - 1$ **do**
6:     Input $x'_{adv}$ and Output $f_\theta(x'_{adv})$
7:     Get Mean Absolute Error Loss $\mathcal{J}(x'_{adv}, y^{true})$ based on $f_\theta(x'_{adv})$ and Eq.(6)
8:     Smooth the gradient of super-pixels patch based on Eq.(3) to obtain the gradient: $\nabla_{\eta_i} \mathcal{J}(x'^i_{adv}, y^{true})$
9:     Input $\nabla_{\eta_i} \mathcal{J}(x'^i_{adv}, y^{true})$, $\mathcal{PCT}$, $\epsilon$ in Algorithm 1 and obtain the output $\xi$
10:     Clip $\eta_{i+1}$ to ensure $||\mathcal{T}(\eta_{i+1})||_\infty \leq \epsilon \cdot m$
11:     $x'^{i+1}_{adv} = \max(0, \min(x + \mathcal{T}(\eta_{i+1}), 1))$
12: **end for**
13: Return Adversarial Samples $x'_{adv}$ and Perturbation $\xi$

---

In the final stage of encoding, we employed Reversible Data Encoding (RDE) technology to embed critical perturbation information into superpixel blocks. Tian et al. Tian (2003) pioneered the RDE technique through difference expansion, a classical method for secret data embedding that amplifies the differences between adjacent pixels. However, conventional RDE methods often introduce distortions in the grayscale representations of images, which are crucial for feature analysis. Therefore, we adopted the Grayscale Invariance RDE (RDE-GI) method proposed by Hou et al. Hou et al. (2018), which utilizes the R and B channels of color images for information embedding while adjusting the pixel values in the G channel to ensure grayscale invariance.

## 3.3 NOISE SERIALIZATION DECODING

In the NSE module, perturbations of varying sensitivity are constructed through superpixel blocks and cleverly embedded into adversarial images using Reversible Data Encoding (RDE) technology.

Thus, in the NSD module, we can utilize the Grayscale Invariance Reversible Data Hiding (RDE-GI) technique to extract hidden information from encrypted private datasets. During the generation of adversarial samples, the perturbation matrix is encoded as a binary information stream, which is then carefully embedded into the adversarial image along with relevant auxiliary data, ensuring the integrity of the embedded information while preserving the adversarial characteristics of the image. When the original image needs to be recovered, the RDE-GI technique can be employed to extract hidden information from the NSE-encoded adversarial image. Given known parameters (such as superpixel size and encoding length), the perturbation matrix can be accurately reconstructed after removal, enabling the recovery of the original image with minimal loss. This innovative approach allows SceneLock to maintain the effectiveness of adversarial perturbations while ensuring process reversibility, thereby achieving robust protection of privacy and sensitive data.

## 3.4 DISCUSSION: ENCODING AND DECODING TIME

Due to the remote deployment requirements of autonomous driving tasks, configuration capabilities, fundamental device requirements, and runtime are crucial factors to consider in the actual data protection process. Therefore, we will primarily discuss the time consumption during the encoding and decoding phases.

**Encoding Time.** In NSE process, we primarily utilize a surrogate model to detect potential target representations in images, which accounts for the main memory overhead. In contrast, superpixel block gradient sensitivity encoding employs binary computation, resulting in minimal time impact. We use 2D detection networks as the surrogate network, operating in validation mode without training, and complete the encoding process in just 10 iterations. Therefore, while the overall encoding process incurs some computational cost, it remains acceptable for current, mature mobile deployments.

**Decoding Time.** The decoding module incurs no GPU overhead and effectively meets the time requirements for extracting specific encrypted data in practical scenarios by combining RDE-GI technology with binary computation. In the subsequent experimental section 6, we further analyze the impact of varying noise levels and resolutions on decoding time.

## 4 EXPERIMENTS

### 4.1 DATASET AND EXPERIMENT SETUP

**nuScenes.** nuScenes Caesar et al. (2020) is a popular dataset for autonomous driving research. It includes 10 object classes, making it an ideal testbed for evaluating semantic learning with massive coarse labels. Given the substantial computational resources required for evaluating the full dataset, we selected the nuScenes-mini dataset to assess adversarial robustness."

**Victim Model.** In SceneLock, the noise encoding flow follows a black-box attack model, which aligns with the application of data to mitigate potential threats from unknown models. In this black-box environment, we selected seven different architectures of pure visual detectors and input images processed with implicitly encoded noise for testing in various 3D detectors, as outlined in Table 2.

**Surrogate Model.** In the noise serialization encoding module, we use lightweight 2D detectors (Fast-RCNN Girshick (2015)) as surrogate models for 3D scene Perception.

**Evaluation.** In our experiments and discussions, we focus on four key dimensions: 3D detection/occupancy attack results, object recognition attack results, image restoration quality, and visual integrity. For 3D detection, we primarily utilize two metrics: Mean Average Precision (mAP) and nuScenes Detection Score (NDS). We evaluate the performance of perturbation noise in NSE and NSD using the Attack Success Rate (ASR), which measures the proportion of targets successfully detected before and after perturbation. Additionally, we employ established benchmarks for image quality assessment to evaluate NSE and NSD, specifically Peak Signal-to-Noise Ratio (PSNR) and Structural Similarity Index (SSIM).

### 4.2 IMPLEMENTATION DETAILS

The parameter settings for SceneLock are as follows: the superpixel unit size is 4x4, the unit perturbation coefficient $\epsilon$ is 4/255, the number of iterations for the surrogate model is 10, and the $\mathcal{PCT}$ is set to 70%. During the attack process, the image resolution is set to 448x448x3. In the NSE module,

Table 1: Differences from Attack Methods

| Method | Surr Model | Attack Type | | | Reversibility |
|---|---|---|---|---|---|
| | | 2D Reg | 3D Det. | 3D Occ | |
| BEV-Attack | 3D | ✓ | ✓ | | |
| Ours | 2D | ✓ | ✓ | ✓ | ✓ |

Table 2: 3D Detector Overview

| 3D Detector | Views | Architecture | Extra signal |
|---|---|---|---|
| PGD-DET | Monocular | CNN | - |
| FCOS3D | Monocular | CNN | - |
| BEVDET | Single | CNN | Depth |
| BEVDepth | Multi | CNN | Depth |
| PETR | Multi | Transformer | - |
| DETR3D | Multi | Transformer | - |
| BEVFormer | Multi | Transformer | Temporal |

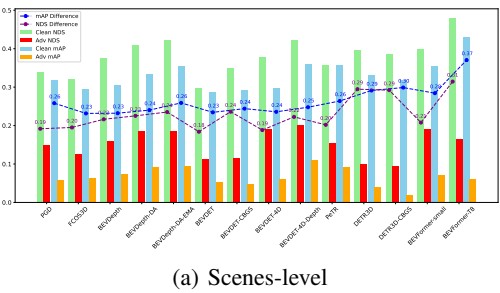

(a) Scenes-level

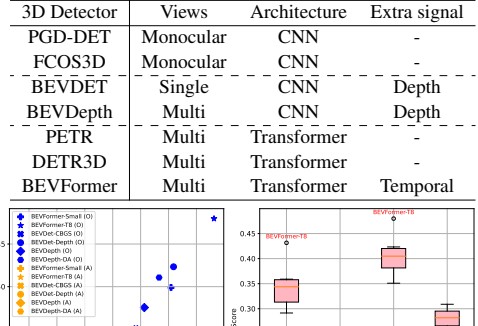

(b) Object-level

Figure 4: Impact of Scene Level on 3D Detector Performance and Influence of Car Object Level on Certain 3D Detectors.

the image resolution is set to 448x448x3 during scene perturbation. In addition to the experiments in Sections 4.3 and 6, we conducted transfer experiments in Appendix B to evaluate the robustness of SceneLock in object recognition tasks. Furthermore, detailed visual analyses are provided in Appendices D and C to further demonstrate the effectiveness of SceneLock in protecting scenes and objects. All experiments were conducted on an NVIDIA GeForce RTX 3090.

### 4.3 3D SCENES RESULTS

In this section, we primarily evaluate the impact of adversarial images encoded by the NSE on the performance of scene perception models.

**Scene-level Results.** In Table 3, we observe that BEVFormer-T8 achieves mAP and NDS scores of 43.15% and 47.98% on the original data, respectively. However, these scores drop to 6.11% and 16.53% on adversarial samples. The trend in Figure 4 (a) reveals that Transformer-based methods (such as BEVFormer, DETR3D, and PETR) perform well on original data but significantly lag behind BEV-based CNN methods when faced with perturbations. This phenomenon suggests that Transformer architectures exhibit less stability than CNNs in learning visual representations.

Furthermore, we tested the 3D occupancy task, as shown in Table 4, which is a pixel-level task more resistant to noise than instance-level detection tasks. However, we observed that the latest Occ method, DHD Wu et al. (2024), achieved only 10.41% mIoU on adversarial samples, with the bicycle class nearly completely disappearing from the scene. Meanwhile, BEVDet and FlashOcc Yu et al. (2023) saw their results drop from 24.23% and 23.98% to 7.51% and 6.47%, respectively. These experimental results demonstrate that our method can effectively disrupt 3D scenes using only a 2D surrogate model, thereby helping to conceal and protect the scene.

**Object-level Results.** Given that cars are the most common and numerous category in road scenes, we conducted noise perturbation experiments specifically on this class, applying noise only to cars in the NSE module. Figure 4 (b) shows the impact on mAP and NDS for various 3D detection methods, with performance degradation varying across models. The box plot indicates that while BEVFormer-T8 performs well on clean data, its effectiveness is significantly reduced on adversarial samples generated by NSE. Table 5 provides more detailed results; although the NDS for cars remains largely unchanged, the 3D bounding box shapes, vehicle trajectories, and speed predictions are significantly disrupted, demonstrating that SceneLock's NSE offers effective protection at the object level.

### 4.4 NOISE SERIALIZATION DECODING RESULTS

**Decoding Time.** In Table 6, we evaluated the impact of different noise levels and resolutions on decoding time. The results indicate that, at the same resolution, varying perturbation noise has a minimal effect on decoding time. In contrast, for the same perturbation level, higher resolutions

Table 3: Performance Comparison of Various 3D Detectors on Clean and Adversarial Scenes-level Data.

| Method | Image Size | CBGS | BEV | Depth | Temporal | Clean NDS | Adv NDS | Clean mAP | Adv mAP |
|---|---|---|---|---|---|---|---|---|---|
| PGD-Det | 1600 × 900 | | | | | 0.3402 | 0.1485 | 0.3173 | 0.0591 |
| PGD-Det-Finetune | 1600 × 900 | | | | | 0.3387 | 0.1386 | 0.3215 | 0.0544 |
| FCOS3D | 1600 × 900 | | | | | 0.3220 | 0.1269 | 0.2948 | 0.0632 |
| FCOS3D-Finetune | 1600 × 900 | | | | | 0.3309 | 0.1313 | 0.3083 | 0.0488 |
| BEVDepth-R50 | 704 × 256 | | ✓ | ✓ | | 0.3755 | 0.1592 | 0.3066 | 0.0744 |
| BEVDepth-R50-DA | 704 × 256 | | ✓ | ✓ | | 0.4107 | 0.1854 | 0.3331 | 0.0929 |
| BEVDepth-R50-DA-EMA | 704 × 256 | | ✓ | ✓ | | 0.4221 | 0.1869 | 0.3541 | 0.0951 |
| BEVDet-R50 | 704 × 256 | | ✓ | | | 0.2980 | 0.1140 | 0.2884 | 0.0540 |
| BEVDet-R50-CBGS | 704 × 256 | ✓ | ✓ | | | 0.3509 | 0.1148 | 0.2915 | 0.0474 |
| BEVDet-R50-4D | 704 × 256 | | ✓ | ✓ | ✓ | 0.3793 | 0.1910 | 0.2967 | 0.0609 |
| BEVDet-R50-4D-Depth | 704 × 256 | | ✓ | ✓ | ✓ | 0.4233 | 0.2007 | 0.3590 | 0.1113 |
| PETR-VovNet | 1600 × 640 | | ✓ | | | 0.3575 | 0.1555 | 0.3571 | 0.0932 |
| DETR3D | 1600 × 900 | | ✓ | | | 0.3954 | 0.1007 | 0.3313 | 0.0400 |
| DETR3D-CBGS | 1600 × 900 | ✓ | ✓ | | | 0.3872 | 0.0951 | 0.3187 | 0.0198 |
| BEVFormer-Small | 1600 × 900 | | ✓ | | | 0.3991 | 0.1913 | 0.3545 | 0.0702 |
| BEVFormer-Base | 1600 × 900 | | ✓ | | | 0.4214 | 0.1601 | 0.3691 | 0.0681 |
| BEVFormer-Base-T1 | 1600 × 640 | | ✓ | | ✓ | 0.3900 | 0.1184 | 0.3430 | 0.0413 |
| BEVFormer-T1 | 1600 × 640 | | ✓ | | ✓ | 0.4132 | 0.1085 | 0.3624 | 0.0337 |
| BEVFormer-T8 | 1600 × 640 | | ✓ | | ✓ | 0.4798 | 0.1653 | 0.4315 | 0.0611 |

Table 4: Performance Comparison of Various 3D Occupancy Methods in Clean and Adversarial Scenes. Gray indicates results in perturbed adversarial scenes.

| Model | mIoU | others | bicycle | bus | car | motorcycle | pedestrian | traffic cone | truck | sidewalk | terrain | manmade | vegetation |
|---|---|---|---|---|---|---|---|---|---|---|---|---|---|
| BEVDet | 24.23 | 27.95 | 2.53 | 29.04 | 38.32 | 8.6 | 11.85 | 4.98 | 22.00 | 41.90 | 31.65 | 42.21 | 29.31 |
| BEVDet | 7.51 | 6.36 | 0 | 6.44 | 17.29 | 0.09 | 1.22 | 0.47 | 5.55 | 13.11 | 4.53 | 13.66 | 12.33 |
| FlashOcc | 23.98 | 25.27 | 1.20 | 25.86 | 38.33 | 12.74 | 11.97 | 5.41 | 18.5 | 42.05 | 36.47 | 41.79 | 26.96 |
| FlashOcc | 6.47 | 0.05 | 0.35 | 6.83 | 6.9 | 0.62 | 1.11 | 0 | 4.95 | 11.13 | 9.28 | 12.28 | 15.2 |
| DHD | 29.12 | 32.87 | 14.22 | 29.23 | 40.66 | 16.57 | 15.77 | 15.94 | 21.8 | 37.7 | 35.03 | 43.20 | 31.86 |
| DHD | 10.41 | 12.95 | 0 | 8.05 | 15.5 | 1.19 | 2.8 | 6.90 | 7.95 | 22.41 | 7.22 | 16.43 | 12.23 |

significantly increase decoding time, primarily due to the increased number of pixels. However, without the use of a GPU, we achieved decoding of noise within one second at a resolution of 448x448, which is acceptable for edge devices.

**Decoding Quality.** In Table 7, we measured the quality of adversarial samples generated by NSE and the quality of the recovered images. The results show that under different noise conditions, the SSIM and PSNR of the recovered images significantly improved. Additionally, higher noise levels made the recovery of the original images increasingly difficult, consistent with the bit capacity limitations discussed in Section 3.2. Excessive perturbation noise led to the loss of high-bit content, further complicating recovery. The reduction of ASR to zero indicates that the object initially missed by BEVFormer in NSE were rediscovered after NSD processing.

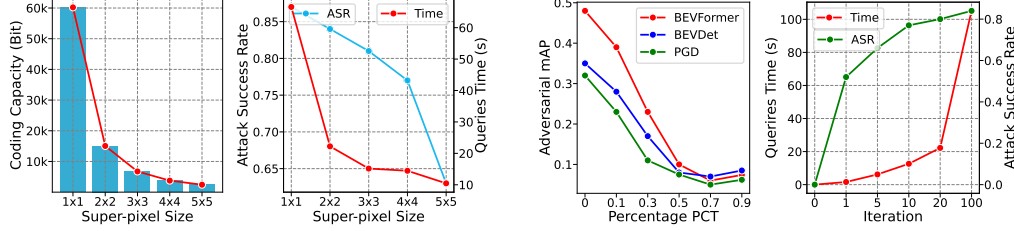

Figure 5: Ablation Analysis of Super-pixel Size, Bit Capacity, and Hyper-parameters $\mathcal{PCT}$ and Iteration.

### 4.5 ABLATION STUDY

This section presents ablation studies on SceneLock to evaluate the effects of various parameters and strategies on its performance, as shown in Figure 5 and Table 8.

**Super-pixel Size.** Increasing the superpixel size reduces the number of image pixels, thereby shortening the encoding length and speeding up perturbation. However, excessive pixel loss degrades

Table 5: Comparison of 3D Detection Performance of Car Object-level under Adversarial Perturbations and Clean Scenes. Gray Represents Results for Perturbed Target Testing.

| Model | NDS | ATE ↓ | ASE ↓ | AOE ↓ | AVE ↓ | AAE ↓ |
|---|---|---|---|---|---|---|
| BEVDET-CBGS | 0.3793 | 0.494 | 0.168 | 0.136 | 0.118 | 0.061 |
| BEVDET-CBGS | 0.2462 | 1.105 | 0.196 | 0.347 | 0.166 | 0.074 |
| BEVDET-Depth | 0.4233 | 0.422 | 0.165 | 0.123 | 0.117 | 0.067 |
| BEVDET-Depth | 0.3089 | 0.768 | 0.198 | 0.339 | 0.170 | 0.104 |
| BEVDepth | 0.4233 | 0.422 | 0.165 | 0.123 | 0.117 | 0.067 |
| BEVDepth | 0.2637 | 0.807 | 0.196 | 0.659 | 0.277 | 0.123 |
| BEVDepth-DA | 0.4107 | 0.476 | 0.168 | 0.149 | 0.159 | 0.086 |
| BEVDepth-DA | 0.2965 | 0.822 | 0.208 | 0.440 | 0.258 | 0.115 |
| BEVFormer-small | 0.3991 | 0.534 | 0.160 | 0.107 | 0.138 | 0.085 |
| BEVFormer-small | 0.2725 | 1.026 | 0.231 | 0.261 | 0.241 | 0.129 |
| BEVFormer-T8 | 0.4798 | 0.356 | 0.171 | 0.080 | 0.108 | 0.071 |
| BEVFormer-T8 | 0.2914 | 0.759 | 0.201 | 0.239 | 0.211 | 0.099 |

Table 6: Comparison of Single Image Recovery Time. Each Result is the Average of Five Repeats.

| Unit Size $\epsilon$ | Recovery Time (s) | | | |
|---|---|---|---|---|
| | 112×112 | 224×224 | 448×448 | 640×640 |
| 3/255 | 0.0082 | 0.0289 | 0.1513 | 0.3741 |
| 4/255 | 0.0089 | 0.0655 | 0.1705 | 0.3852 |
| 5/255 | 0.0093 | 0.0667 | 0.3440 | 0.3911 |

Table 7: Comparison of Adversarial Image and Reconstructed Image Quality, with Statistics on Attack Success Rate.

| Unit Size $\epsilon$ | Adversarial | | | Recover | | |
|---|---|---|---|---|---|---|
| | SSIM ↑ | PSNR ↑ | ASR | SSIM ↑ | PSNR ↑ | ASR |
| 3/255 | 0.7350 | 74.63 | 65.79 | 0.9858 | 79.64 | 0 |
| 4/255 | 0.6519 | 72.94 | 74.12 | 0.9475 | 77.86 | 0 |
| 5/255 | 0.5722 | 71.50 | 85.84 | 0.8692 | 74.81 | 0 |

Table 8: Performance of mAP under Different Settings of the NSE Module in Ablation Analysis.

| 3D Model | Super-pixel Size | | | | Iterations $\mathcal{I}$ | | | Percentage $\mathcal{PCT}$ | | |
|---|---|---|---|---|---|---|---|---|---|---|
| | 2 ×2 | 3×3 | 4×4 | 5×5 | $\mathcal{I} = 5$ | $\mathcal{I} = 10$ | $\mathcal{I} = 20$ | $\mathcal{PCT} = 0.3$ | $\mathcal{PCT} = 0.5$ | $\mathcal{PCT} = 0.7$ |
| BEVDet-R50 | 0.0317 | 0.0429 | 0.0540 | 0.0881 | 0.1244 | 0.0540 | 0.0498 | 0.1533 | 0.0946 | 0.0540 |
| BEVDepth-R50 | 0.0389 | 0.0455 | 0.0740 | 0.1006 | 0.1301 | 0.0740 | 0.0685 | 0.1442 | 0.1105 | 0.0740 |
| BEVFormer-Small | 0.0320 | 0.0431 | 0.0702 | 0.1252 | 0.1123 | 0.0702 | 0.0626 | 0.1772 | 0.1013 | 0.0702 |

perturbation performance. Conversely, with a 1x1 superpixel size, while performance is optimal, the bit capacity significantly increases, adversely affecting query speed. Considering both speed and performance, a 4x4 superpixel size is the optimal choice.

**Iteration and Percentage.** Significant performance differences are observed during the first 20 perturbation iterations; beyond this point, performance improvements become marginal, while perturbation time increases significantly. Additionally, a higher $\mathcal{PCT}$ percentage of top-ranked entries in the gradient score matrix enhances the model's attack efficiency. However, as this percentage approaches 0.9, there is a risk of encountering a gradient trap, resulting in misguidance from ineffective gradient flows.

## 5 CONCLUSION

This paper presents a novel reversible adversarial learning framework, SceneLock, to protect camera-based autonomous driving data from unauthorized use. By embedding adversarial perturbations through Noise Serialization Encoding (NSE), we degrade the data quality for unauthorized models while allowing legitimate users to restore the original data via a Noise Serialization Decoding (NSD) module. Experimental results demonstrate that this method significantly impairs the performance of unauthorized scene perception and recognition tasks while maintaining data integrity for authorized applications. The reversible design ensures minimal data loss during the recovery process, achieving a balance between data protection and accessibility. Future work will extend this method to multimodal data protection and enhance computational efficiency for real-time deployment. This study underscores the potential of reversible adversarial techniques to bolster data security in AI systems, particularly in the autonomous driving domain.

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

# A  BACKGROUND

As discussed in Section 3.1, we will supplement the fundamental background principles of Steganography to enhance the reader's understanding of the paper. Initially, steganography was primarily used in cryptography and information security, but it has since been introduced into image encryption Cheddad et al. (2010). Therefore, we will briefly introduce the fundamental principles of steganography in images and its theoretical basis.

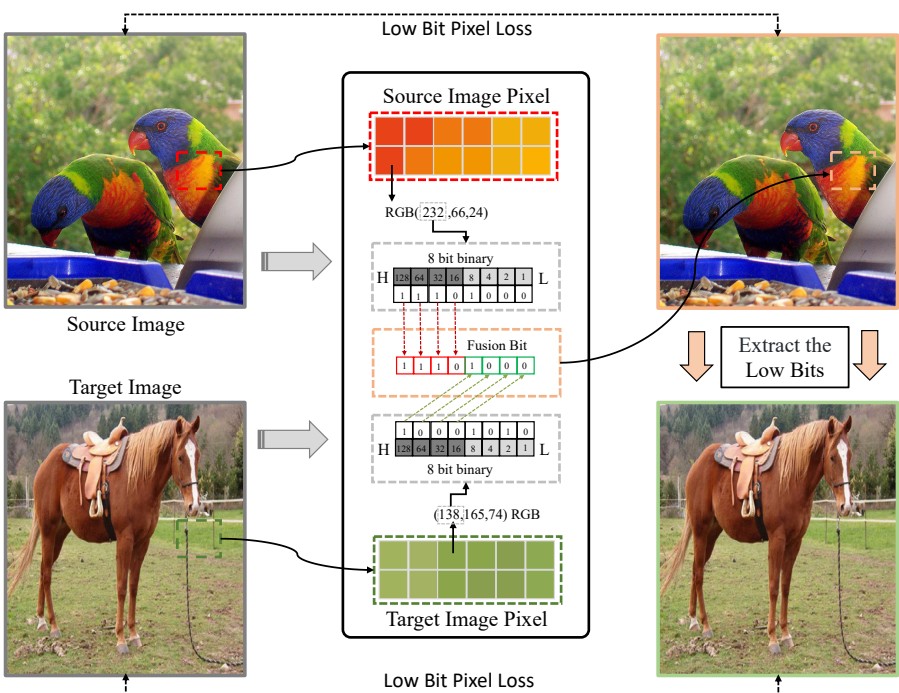

Figure 6: The Overall Framework of Steganography in Image Processing.

A fundamental consensus in computer vision is that the way computers store images differs significantly from human visual perception. Computers store images by converting the RGB values of each pixel into 8-bit binary codes. Additionally, the storage system typically employs a little-endian format, where lower-order data resides at lower memory addresses. It is particularly noteworthy that *human eye is not sensitive to the content of the lower bit positions in pixels*, as the values in higher bit positions dominate the image. Based on this theoretical foundation, we can embed information from a target image into the low bit positions of a source image for the purpose of hiding, as illustrated in Figure 6. Convert each pixel of the source and target images into 8-bit binary format. Then, write the high-order bits of the target image into the source image, replacing its low-order bits. This method allows for the fusion of each pixel, resulting in a new image that visually resembles the source image. The method for extracting hidden target images involves retrieving all low-order bit data from the newly synthesized image and rewriting it into the high-order bits to recover the target image. However, it is important to note that the recovery in steganography does not imply the possibility of achieving a completely lossless image reconstruction. In reality, this process is conducted at the expense of low-order bit information. Nonetheless, relative to high-order bits (such as 16, 32, 64, 128), the information loss associated with low-order bits (e.g., 1, 2, 4, 8) is typically negligible.

**Novelty of SceneLock.** We are the first to combine steganography with adversarial attacks in deep learning for the protection of autonomous driving scenarios. We innovatively embed perturbation noise into the source image, successfully concealing the noise while preserving the semantic information of the adversarially perturbed image.

**Limitations of Steganography.** In practical applications of steganography, a critical issue that requires special attention is bit overflow, which may lead to irretrievable images. When the majority of bits in an 8-bit binary data are filled with 1s, the added perturbation noise can potentially trigger an 8-bit overflow. Consequently, our research addresses this problem and implements appropriate

overflow checks to prevent content degradation. However, such measures may restrict the ability to perturb some key content, thus acknowledging that the current bit capacity imposes certain limitations on the effectiveness of perturbations.

# B  MORE ROBUSTNESS EXPERIMENTS

**Dataset.** We selected ILSVRC2012 Russakovsky et al. (2015) as the dataset for our recognition experiments. This dataset is widely utilized in the field of deep learning, characterized by its substantial scale and significant impact. It encompasses 1,000 distinct categories, with each image accurately labeled. We primarily employ mean Average Precision (mAP) and Attack Success Rate (ASR) as evaluation metrics.

**Victim Model.** For object recognit ion, we employed performance evaluation methods based on multiple architectures, including CNN (ResNet He et al. (2016), VGG Simonyan & Zisserman (2014), Inc-V3 Szegedy et al. (2016) and Dense Huang et al. (2017)), Transformer (Swin Transformer Liu et al. (2021) and ViT Dosovitskiy (2020)), and CLIP Radford et al. (2021).

**Surrogate Model.** For object recognition, we employ the used ResNet He et al. (2016) and ViT Dosovitskiy (2020) as a surrogate model.

Table 9: Performance of NSE-generated adversarial results across different methods. † denotes ResNet50 as the surrogate model, while ‡ denotes ViT as the surrogate model.

| Model | Clean | Adv † | Adv ‡ |
|---|---|---|---|
| Inc-V3 | 0.669 | 0.033 | 0.170 |
| VGG19 | 0.709 | 0.045 | 0.167 |
| Dense121 | 0.729 | 0.110 | 0.173 |
| ResNet34 | 0.710 | 0.125 | 0.011 |
| ResNet152 | 0.773 | 0.056 | 0.243 |
| ViT-b-16 | 0.791 | 0.567 | 0.263 |
| Swin-s | 0.820 | 0.643 | 0.485 |
| CLIP-vit-b/32 | 0.570 | 0.327 | 0.318 |

Table 10: Image quality results under different noise perturbations, all based on ResNet as the surrogate model.

| Noise Size | Adversarial | | | Recover | | |
|---|---|---|---|---|---|---|
| | SSIM ↑ | PSNR ↑ | ASR | SSIM ↑ | PSNR ↑ | ASR |
| 3/255 | 0.8526 | 75.84 | 79.3 | 0.9947 | 79.78 | 0 |
| 4/255 | 0.7987 | 74.50 | 83.5 | 0.9821 | 78.33 | 0 |
| 5/255 | 0.7513 | 73.35 | 92.9 | 0.9585 | 75.59 | 0 |

**Object Recognition Results.** In this study, we applied our proposed data protection method Scene-Lock (NSE and NSD) to various categories of data and utilized models of different structures and scales for recognition testing. Table 9 demonstrates that when perturbations are applied using CNN-based surrogate models, the performance of CNN methods significantly decreases, whereas the performance loss in Transformer-based methods is less pronounced. Conversely, perturbations with Transformer-based surrogate models lead to a significant decline in recognition rates for most models. This indicates that the Transformer structure is more stable than CNN for single image recognition. Furthermore, these results suggest that our proposed SceneLock method can flexibly employ different surrogate models to enhance the success rate of perturbations.

**Image Quality.** In Table 10, we evaluate the image quality of adversarial and recovered samples processed through the NSE and NSD modules. It can be observed that object recognition follows a similar pattern to scene perception: as the perturbation intensity increases, the probability of high-bit information loss rises, making image quality restoration more challenging. Furthermore, this demonstrates that SceneLock can effectively disrupt semantic content with appropriate perturbation intensity while achieving excellent image recovery.

# C  VISUALIZATION ANALYSIS OF SCENES

**Scene Protection Visualization.** In this section, we provide a visual explanation of the NSE and NSD modules. Figure 9.(a) presents a gradient heatmap normalization visualization of a specific object within the scene, with the red bounding box indicating the selected object to be concealed.

In the adversarial sample, it can be observed that the gradient information within the red bounding box disappears in the GT scene, effectively removing the object from the scene. In Figure 9.(b), we encoded the entire scene, and the heatmap shows that we successfully concealed objects within the scene under the perception model. This demonstrates that SceneLock can effectively protect both individual objects and entire scenes.

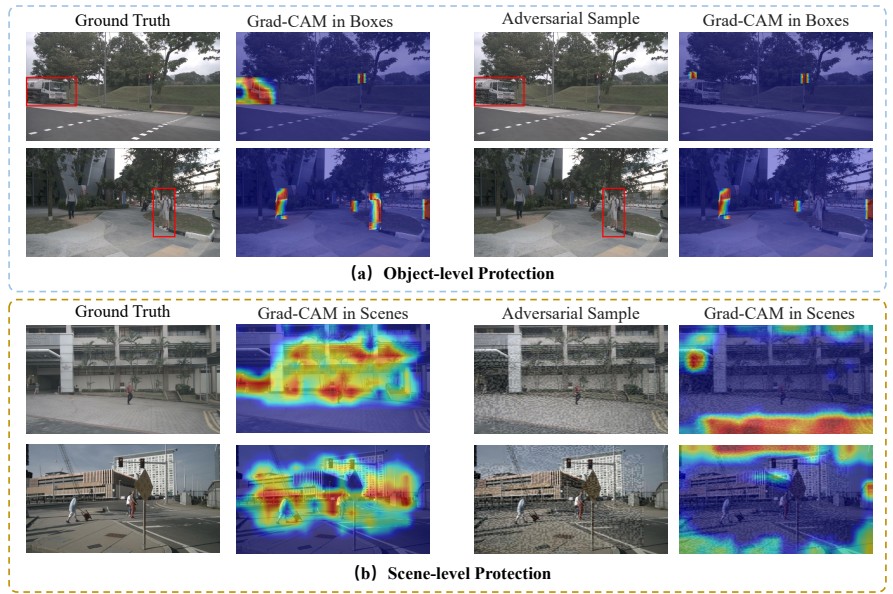

Figure 7: Gradient heatmap visualization. (a) NSE-encoded noise achieves object-level target concealment. (b) NSE-encoded noise achieves scene-level concealment.

**Scene Image Quality Visualization.** In Figure 8, we visualize the results of different noise intensities. As the noise level increases from 3/255 to 5/255, the watermark mask on the adversarial images becomes more prominent, indicating stronger scene perturbation and greater protection coverage. However, this leads to a decline in the quality of the recovered images. When the noise is set to 3/255, the SSIM value is close to that of the original clean image, whereas at 5/255, both SSIM and PSNR exhibit a significant drop. This demonstrates that SceneLock can achieve effective scene protection by selecting an appropriate noise intensity while still enabling the recovery of high-quality images.

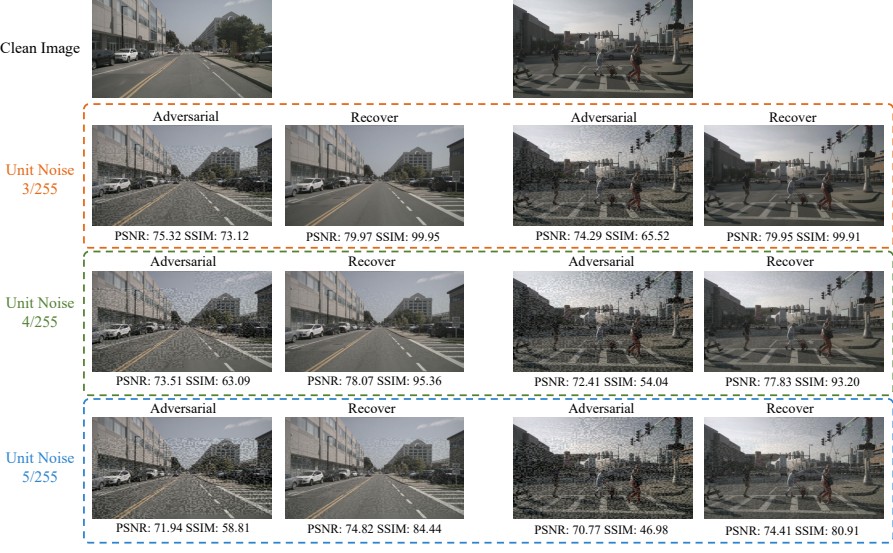

Figure 8: Visualization of images and their quality under different noise intensities.

**Scene Perception Visualization.** SceneLock aims to prevent pervasive perception models from infringing on privacy within a scene, making visual assessment of current mainstream 3D perception tasks crucial. As shown in Figures 9 and 10, the adversarial data generated by SceneLock through the NSE module performs poorly in 3D detection tasks, with most objects remaining undetected. In dense semantic tasks such as Occ, vehicles and pedestrians in the DHD-NSE scene are also absent, indicating that SceneLock effectively protects scenes against dense semantic perception models. Furthermore, after image recovery through NSD, the performance of both the 3D detector and the Occ prediction model returns to normal, successfully achieving scene perception.

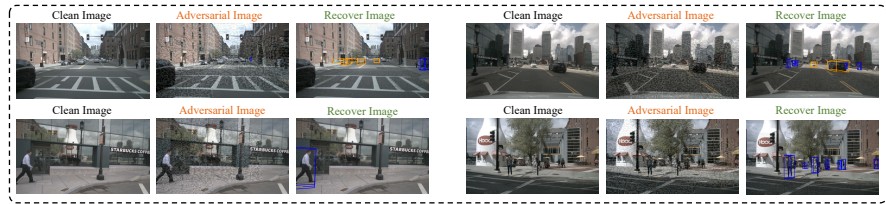

Figure 9: Visualization of 3D detection results in SceneLock encoding and decoding using BEVDet.

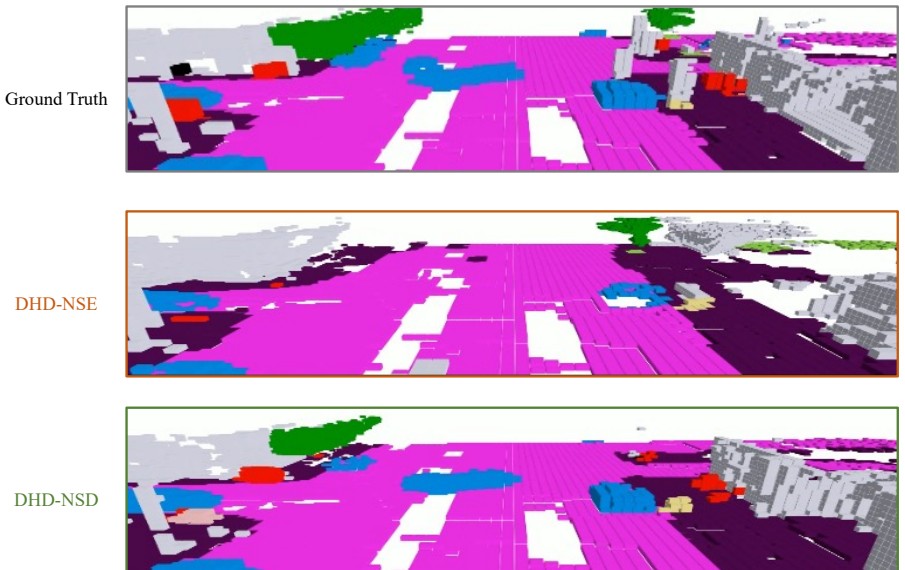

Figure 10: Visualization of 3D occupancy prediction results in SceneLock encoding and decoding using DHD. Blue voxels represent car.

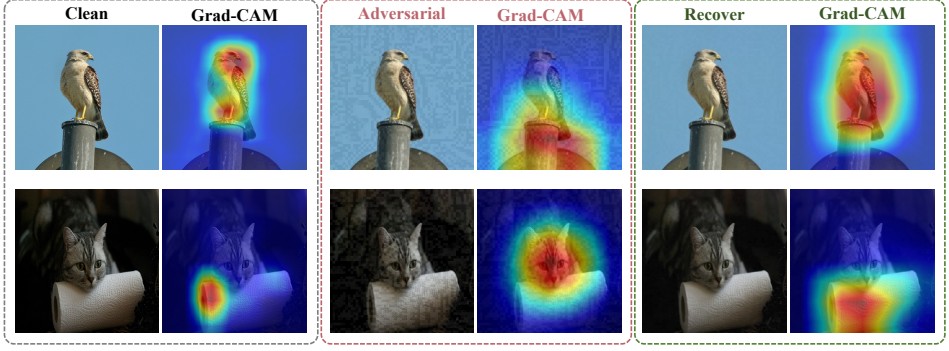

Figure 11: Grad-CAM Visualization of Clean, Adversarial, and Recovered Data.

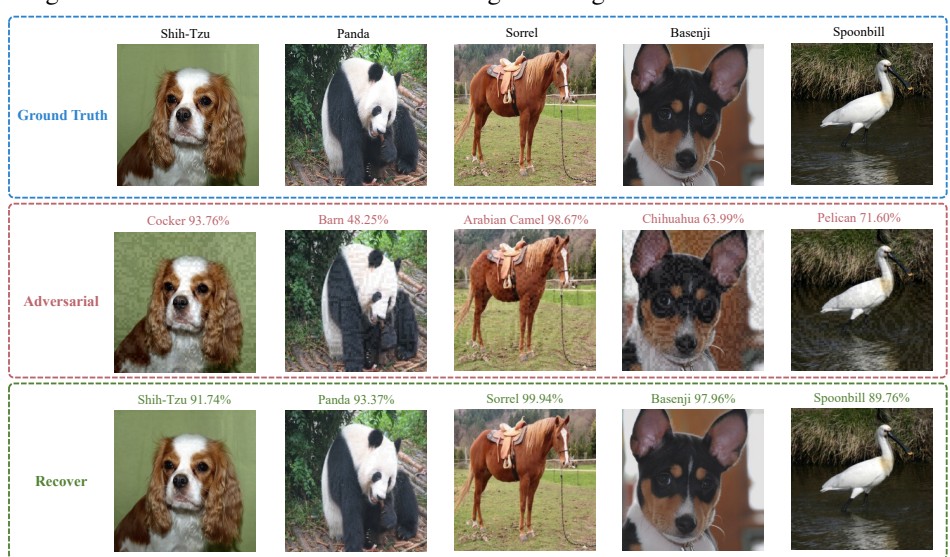

Figure 12: Grad-CAM Visualization of Images During Perturbation Iteration Process.

Figure 13: Visualization of Recognition Results Based on ResNet Model.

# D  VISUALIZATION ANALYSIS OF OBJECT

**Object Protection Visualization.** We extend the data protection features of SceneLock to non-scene data, further validating its effectiveness as a general framework. As illustrated in Figure 11, we input clean images, adversarial images, and recovered images into the ResNet network for heatmap visualization. The results indicate that the model effectively focuses on the correct features and makes accurate class predictions for clean images. In contrast, Grad-CAM fails to capture the correct class information in adversarial images, redirecting attention to other semantic areas. After recovery, both Grad-CAM and the model correctly identify the target again. Additionally, Figure 12 provides a detailed visualization of the model's attention changes throughout the iterative process. This sequence demonstrates that SceneLock can effectively protect classification tasks at the semantic level, preventing malicious extraction or misuse of images.

**Visualization of Object Prediction Results.** We visualized the prediction results for both accessible (locally loadable models) and inaccessible (commercial API interfaces) models. As shown in Figure 13, when ResNet50 is used as a surrogate model for data protection, adversarial images tend to cause incorrect predictions, while the recovered images are correctly classified with high confidence in the GT labels. In Figure 14, we tested recognition using commercial APIs, where training models are typically difficult to obtain, making attacks more challenging. The results demonstrate that SceneLock-protected data effectively misled the recognition models in these commercial APIs as well. In summary, this further validates the strong performance of SceneLock in cross-model protection.

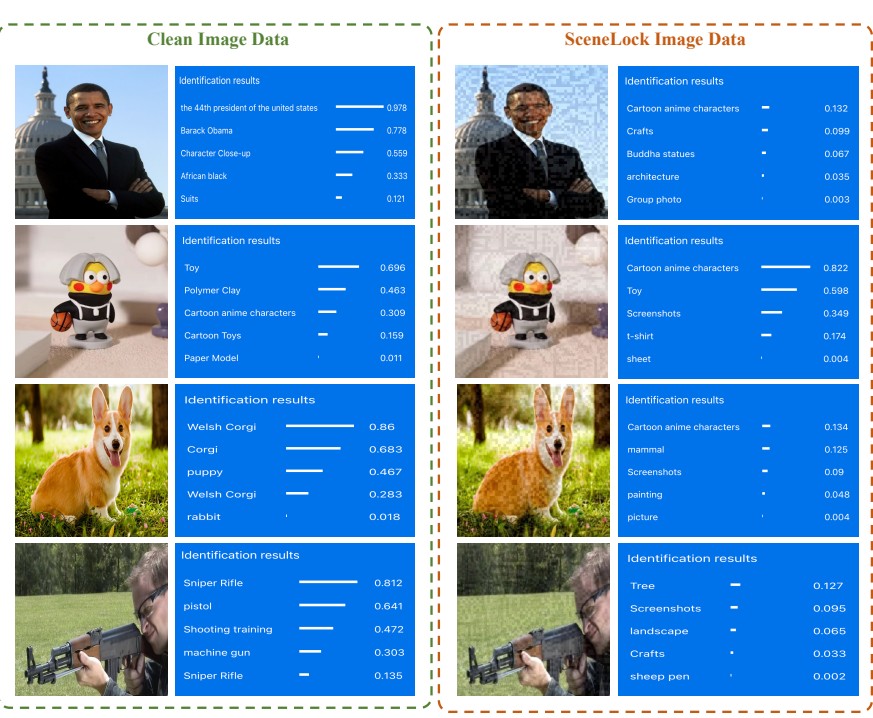

Figure 14: Visualization of Recognition Results from Commercial API Model. The blue image content represents the result returned by the server.