# OpenReview forum: "SceneLock: Reversible Adversarial Learning for Camera-Based Autonomous Driving Protection"
_ICLR.cc/2025/Conference — ICLR 2025 Conference Withdrawn Submission_

### Official Review · Reviewer_nYyC · 2024-10-30

**Soundness:** 3
**Presentation:** 1
**Contribution:** 2
**Rating:** 3
**Confidence:** 3

**Summary:**

This work propose a novel reversible adversarial learning framework, which is termed SceneLock, for the protection of camera-based autonomous driving scenes. SceneLock conducts adversarial perturbations through a carefully designed Noise Serialization Encoding module (NSE), which significantly degrades image quality and renders the data ineffective for unauthorized artificial intelligence models and manual annotation. To ensure legitimate access remains unaffected, SceneLock integrate advanced image steganography to embed perturbation values within the images. Furthermore, authorized users can extract these values using appropriate decryption tools
through the Noise Serialization Decoding module (NSD) to restore the original high-quality images.

**Strengths:**

This paper proposes SceneLock framework, which protects data by countering disturbance and guarantees the normal authorization parsing of images by integrating image steganography. This framework can effectively protect against unauthorized access and misuse during large-scale image transmission.

**Weaknesses:**

Although there is some innovation in this work, the overall organizational structure and writing logic of the article are very poor. It is suggested to re-examine the problem and think about the writing Angle.

**Questions:**

1. The essence of this paper lies in steganography and encryption of images to safeguard privacy during transmission, preventing unauthorized access. Image security is a crucial area of research in its own right. However, the application of these concepts to camera-based autonomous driving raises important questions: Do such scenarios introduce unique features or research opportunities for image steganography? I believe the author’s perspective could be improved, and I recommend a clearer and more focused approach in the revision.

2. In the introduction, while the author summarizes the innovation points, there is a lack of a brief overview of the current research landscape regarding this issue. Specifically, it would be beneficial to highlight existing research gaps and the challenges addressed by the solutions proposed in this paper.

3. In the related work section, the authors should concentrate on existing research pertinent to the problem at hand, such as studies on image/data perturbation and the main techniques being employed, like image steganography. The introduction of 3D camera perception research appears disconnected from the core problem being addressed and lacks clear significance.

4. The overview section (3.1) should not serve as a roadmap. This section should provide a structured outline of the overall framework, workflow, and mechanisms employed in the study.

5. Most references cited by the author are dated prior to 2020, resulting in a significant lack of recent studies. Incorporating more contemporary references would strengthen the paper's relevance and authority.

---

### Official Review · Reviewer_W1VN · 2024-11-01

**Soundness:** 2
**Presentation:** 2
**Contribution:** 2
**Rating:** 3
**Confidence:** 4

**Summary:**

The paper presents SceneLock to protect deep neural network 3D object detectors from unauthorized access and misuse. SceneLock employs a reversible adversarial framework that integrates a Noise Serialization Encoding (NSE) module to apply adversarial perturbations. NSE can degrade image quality and render the data ineffective for unauthorized use. To maintain access for legitimate users, the authors incorporate image steganography to embed perturbation values within the images, allowing authorized users to restore the original quality through a Noise Serialization Decoding (NSD) module. The experimental results demonstrate that SceneLock effectively safeguards data integrity while preserving its availability for authorized purposes.

**Strengths:**

1.	The SceneLock framework introduces a novel dual-layer protection mechanism for protecting sensitive autonomous driving data.
2.	The authors provide thorough experimental results to demonstrate the effectiveness of SceneLock.

**Weaknesses:**

1.	The threat model is not convincing. Can malicious actors gain unauthorized access to the autonomous driving data? Existing techniques such as access control could be directly implemented to address unauthorized access issues, which might be simpler and more effective.
2.	While the authors aim to protect intellectual property, remote deployment, and privacy, the proposed method does not directly tackle IP protection. The approach might offer some level of privacy protection, but this assertion feels subjective due to the absence of quantitative privacy assessments.
3.	The data hiding approach for embedding and extracting restoration messages seems unnecessary if only authorized users can access the data. In practical scenarios, encryption with secure transmission and secret key channels could achieve similar outcomes more efficiently.
4.	The adversarial learning approach with data hiding has similarities to other works in unlearnable examples with data hiding. However, I cannot find the advantage of this paper. And there is a lack of comparative evaluation against these existing methods.

**Questions:**

Please refer to the weaknesses.

---

### Official Review · Reviewer_pgms · 2024-11-01

**Soundness:** 1
**Presentation:** 2
**Contribution:** 1
**Rating:** 3
**Confidence:** 4

**Summary:**

This work proposes a framework that applies adversarial perturbations to protect image information from unauthorized use. Specifically, the input image is passed through an encoding module that degrades its quality by applying adversarial perturbations, making the perturbed image difficult to be processed by unexpected DNNs. A decoding model, intended for use by authorized parties, then restores the original image from the perturbed version. Experiments were conducted to evaluate the ability of unauthorized models to process the perturbed images.

**Strengths:**

- The objective of the work addresses an interesting high-level problem related to protecting only authorized use.

- Results demonstrate that the perturbations effectively reduce the performance of unauthorized users across various computer vision tasks.

**Weaknesses:**

- Unclear connection with steganography. There is a lack of clarity regarding the connection between the approached adopted in this work and steganography. The authors should revise the manuscript to clarify this point. Specifically, it is unclear if steganography is the appropriate term, as the goal appears to be limiting image processing by sharing the decoder only with authorized users, without concealing the image from human view.

- Unclear threat model. The authors should further clarify the rationale for using adversarial perturbations for data protection. The introduction lacks citations and clear motivation for this approach. For example, one might question the necessity of this method: why not use traditional encryption instead?

- Limited discussion of related literature.  Firstly, the extensive discussion of camera-based 3D perception paragraph in the related works appears to be unrelated. Secondly, while adversarial perturbations and vision models in driving are mentioned, other relevant works that could support this strategy and emphasize the importance of efficiency in this context are missing, along with a clear threat model.

- Absence of analysis against potential image extraction attacks. Given the need for a clear threat model, if the goal is to make the dataset publicly available but unusable by unauthorized users, the methodology should consider potential denoising strategies that attackers might use to remove the noise. This critical aspect has not been discussed, yet it is essential for assessing the work’s quality. Since this approach is proposed as a potential defense mechanism, the authors should explore and test it against possible attacks, as the lack of such evaluations raises doubts about the pipeline’s quality and robustness. For example, a straightforward test could involve attackers applying denoising or image-reconstruction strategies to the perturbed images, using possible unsupervised and supervised techniques.

- [minor issues] In section 3, "RDE" is used both for Reversible Data Embedding and Reversible Data Encoding.

**Questions:**

-It is unclear why superpixels were addressed in this context. Is efficiency an issue here? Applying noise and denoising could be done offline by both sender and receiver, where effiency is not one of the main problem to address. The authors mention this but do not provide a clear motivation or supporting citations. The authors should better discuss this point and justify why they chose superpixels over other approaches.

- Regarding one of the main issues in this work, do the authors believe that potential defense strategies could be applied against this approach? What is the intended threat model in this context?

- Could the authors clarify the rationale for using adversarial perturbation as a data protection tool? Additionally, where is the connection to steganography, since the goal here is not to hide information but to make it less accessible to unauthorized users?

---

### Official Review · Reviewer_qzrm · 2024-11-02

**Soundness:** 3
**Presentation:** 3
**Contribution:** 2
**Rating:** 5
**Confidence:** 5

**Summary:**

This paper proposes a reversible adversarial learning framework that can generate reversable adversarial perturbations onto dataset images to prevent unauthorized used from using the dataset in model training. Authorized users can use a decoding approach to restore the original dataset. Authors have evaluated their approach regarding attack performance and restoration quality.

**Strengths:**

1. Authors propose a novel adversarial perturbation encoding and decoding approach to protect the dataset from unauthorized usage and privacy leakage.
2. The adversarial perturbations demonstrate good attack performance on both 2D and 3D perception models.
3. The presentation of the paper is clear and easy to follow.

**Weaknesses:**

1. The motivation or threat model of the paper is not convincing. If the dataset to be protected is not suitable for public usage, the dataset owner can encrypt the whole dataset and only share password to authorized users. The motivation to poison the dataset before publishing is not clear.
2. The adversarial noise on the image seems easy to be mitigated by smoothing or other preprocessing techniques. Authors could evaluate the resistance of the perturbations under defensive techniques.
3. Authors evaluated the attack performance of the generated perturbations on various models. A question is that whether those models trained on original clean dataset or the encoded dataset? If those models are trained on original dataset, how would those models perform if they are trained on the encoded dataset?

**Questions:**

Please see the weaknesses above.

---

### Official Review · Reviewer_NgNE · 2024-11-04

**Soundness:** 3
**Presentation:** 2
**Contribution:** 2
**Rating:** 5
**Confidence:** 4

**Summary:**

This paper addresses privacy protection for autonomous driving data using adversarial attacks. The authors propose a framework called SceneLock, which consists of two modules, NSE and NSD. NSE is responsible for transforming original images into adversarial counterparts, making them undetectable and unrecognizable by downstream models. NSD converts these adversarial counterparts back into benign images to allow authorized use. The authors evaluate the effectiveness of the detector under a black-box setting to demonstrate the protection capability.

**Strengths:**

1. This paper addresses an important issue, namely privacy protection for autonomous driving data, which is beneficial for the deployment and development of autonomous driving technology.
2. The reversible adversarial learning framework proposed in this paper is straightforward and has practical applicability.

**Weaknesses:**

1. Limited novelty. This paper introduces adversarial perturbations into images to thwart malicious data users, then uses image steganography techniques to remove the perturbations. Both adversarial examples and image steganography are well-established techniques, so the novelty and contribution of this work are limited.
2. Experiments. The final step of NSE uses Reversible Data Encoding (RDE) technology to embed perturbation information into superpixel blocks. It is unclear whether RDE affects the performance of the adversarial perturbations.
3. Discussion. Image steganography techniques are easily affected by image editing. If this proposed solution undergoes image editing, such as JPEG compression, it might prevent the extraction of perturbation information from adversarial images, making image restoration impossible. This point lacks discussion.
4. Discussion. How robust are the protected images generated by NSE against image denoising algorithms? Would adding suitable denoising algorithms negate the perturbations?
5. Experiments. This paper claims that the degraded image quality hinders manual annotation efforts by unauthorized parties, but there is no experimental evidence to support this claim.

**Questions:**

1. This paper investigates privacy protection for autonomous driving data. What challenges does this field face?
2. Punctuation is missing in the equations. Please add it.
3. The multiplication symbol is not standardized in the paper. For example, "4x4" in Line 322 should be revised to "4\times4". Similar issues appear elsewhere in the paper.
4. Citations are inconsistent. The authors use only the \cite command, but in most cases, \citep would be more appropriate. Additionally, author names can be omitted when using \cite, such as in Line 263.
5. There is an extra quotation mark at the end of the paragraph introducing the dataset.

---

### Note · Authors · 2024-11-20

I have read and agree with the venue's withdrawal policy on behalf of myself and my co-authors.